# Turbulent erosion of a subducting intrusion in the Western Mediterranean Sea

Giovanni Testa[1], Mathieu Dever[2,3], Mara Freilich[4], Amala Mahadevan[2], T. M. Shaun Johnston[5], Lorenzo Pasculli[1,6], Francesco M. Falcieri[1]

[1] Institute of Marine Sciences, Italian National Research Council (CNR-ISMAR), Venice, Italy.
[2] Woods Hole Oceanographic Institution, Woods Hole, 02543, MA, USA
[3] RBR, Ottawa, Canada
[4] Brown University, Providence, RI, USA
[5] Scripps Institution of Oceanography, University of California, San Diego, La Jolla, CA, USA
[6] Department of Environmental Sciences, Informatics and Statistics, University Ca' Foscari of Venice, Via Torino 155, 30172 Mestre, Italy

*Correspondence to*: Giovanni Testa (giovanni.testa@ve.ismar.cnr.it)

**Abstract.** Frontal zones within the Western Alboran Gyre (WAG) are characterized by a density gradient resulting from the convergence of Atlantic and Mediterranean waters. Subduction along isopycnals at the WAG periphery can play a crucial role in upper ocean ventilation and influences its stratification and biogeochemical cycles. In 2019, physical parameters (comprising temperature, salinity, turbulent kinetic energy dissipation rates) and biogeochemical data (oxygen and chlorophyll-a) profiles were collected in transects along the northern edge of the WAG. Several intrusions of subducted water with elevated oxygen, chlorophyll-a and spice anomaly were identified towards the center of the anticyclone. These features had elevated kinetic energy dissipation rates on both their upper and lower boundaries. Analysis of the turbulent fluxes involving heat, salt, oxygen, and chlorophyll-a demonstrated a net flux of physical and biogeochemical properties from the intrusions to the surrounding ocean. Either the turbulent or diffusive convection mixing contributed to the observed dilution of the intrusion. Other factors (e.g., water column density stability, variability of the photic layer depth, and organic matter degradation) likely played a role in these dynamics. Enhanced comprehension of the persistence and extent of these features might lead to an improved quantitative parametrization of relevant physical and biogeochemical properties involved in subduction within the study zone.

## 1 Introduction

The Mediterranean Sea is characterized by a shallow circulation cell and a complex upper-layer circulation featuring numerous quasi-permanent eddies and fronts (Tanhua et al., 2013; Capó et al., 2019; Barral et al., 2021; Bonaduce et al., 2021; Zarokanellos et al., 2022; Sánchez-Garrido and Nadal, 2022). The main 12 Mediterranean thermal fronts were listed by Belkin and Cornillon (2007), whereas a recent work by Sudre et al. (2023) captured an even more complex scenario. Specifically, frontal zones in the Alboran Sea (Western Mediterranean basin) are characterized by a density gradient resulting from the

convergence of Atlantic and Mediterranean waters (Fedele et al., 2022; Garcia-Jove et al., 2022). The Atlantic jet strongly
influences the formation of two large-scale anticyclonic gyres within the Alboran Sea (the Eastern and Western Alboran Gyres,
WAG; **Fig. 1A**) with a smaller cyclonic gyre typically situated in between (Brett et al., 2020; Sala et al., 2022; Sánchez-Garrido
and Nadal, 2022).
Ocean subduction, defined as the physical transfer of water from the mixed layer into the ocean interior (Williams,
2001), plays a pivotal role in upper-ocean ventilation and stratification. It also exerts a profound influence on biogeochemical
cycles, thereby contributing to the export of greenhouse gases and the vertical transport of organic carbon (Omand et al., 2015;
Olita et al., 2017; Stukel et al., 2017; Ruiz et al., 2019; Zarokanellos et al., 2022). The vertical component of ocean current
velocity is typically much smaller than its horizontal counterparts, but areas characterized by meandering frontal features
associated with mesoscale eddies are expected to exhibit elevated subduction rates (van Haren et al., 2006). Indeed, vertical
velocities of up to 55 m d$^{-1}$ have been observed in the Western Alboran Sea front (Capó and McWilliams, 2022; Garcia-Jove
et al., 2022; Rudnick et al., 2022), and net submesoscale subduction rate has been estimated at 0.3 m day$^{-1}$ (Freilich and
Mahadevan, 2021). Mesoscale turbulence contains more energy than submesoscale patterns (Storer et al., 2022), although
submesoscale features can generate larger vertical velocities than mesoscale structures within frontal zones (Mahadevan, 2016;
Ruiz et al., 2019). The relationship between submesoscale velocity and mixing within the boundary layer has been explored
in prior studies under conditions of turbulent thermal wind balance (Crowe and Taylor, 2018; McWilliams, 2021) and
symmetric instabilities (Thomas et al., 2013; Bachman et al., 2017; Zhou et al., 2022). However, so far there has been limited
research that specifically identifies occurrences of quasi-balanced subsurface vertical velocity and examines how mixing
responds to such instances.
Vertical motion at fronts is driven by frontogenesis, instability processes, nonlinear Ekman effects, and
filamentogenesis (Klein and Lapeyre, 2009; Mahadevan, 2016; Mahadevan et al., 2020a; McWilliams, 2021; Capó and
McWilliams, 2022; Garcia-Jove et al., 2022). Instabilities have also been identified as a key source of turbulence and energy
dissipation at oceanic fronts (D'Asaro et al., 2011; Carpenter et al., 2020; McWilliams, 2021). Subsurface intrusions carry
physical (temperature and salinity) and biogeochemical properties (oxygen and chlorophyll-a) characteristic of the surface
mixed layer along isopycnals and extend downward and laterally. Intrusions are often identified because of the co-occurrence
of subsurface maxima in oxygen, particulate organic carbon with anomalous temperature and salinity properties (i.e., spice;
Omand et al., 2015). Intermittent intrusions subducting along the outer periphery of mesoscale and submesoscale structures
have previously been identified (Johnston et al., 2011; Llort et al., 2018; Chapman et al., 2020; Johnson and Omand, 2021;
Chen et al., 2021; Capó and McWilliams, 2022; Freilich et al., 2024). This study measures the turbulent erosion of a subducting
intrusion at fronts within the Western Alboran Gyre, a major mesoscale feature in the western Mediterranean Sea with a Rossby
number of 0.08. Data were collected in the framework of the Coherent Lagrangian Pathways from the Surface Ocean to Interior
(CALYPSO) project onboard the R/V *Pourquoi Pas?*, that aimed to examine subduction features in close proximity to the
unstable front that developed along the northern edge of the WAG (Mahadevan et al., 2020).
Previous studies have investigated turbulence data collected with microstructure probes in both the surface (Cuypers
et al., 2012; Forryan et al., 2012; Vladoiu et al., 2021) and deep (Ferron et al., 2021; van Haren, 2023) regions of the Western
Mediterranean Sea. However, this work represents the first comprehensive investigation of turbulence in a context of
mesoscale-submesoscale subduction at frontal zones within the WAG. This paper begins with a comprehensive description of
water column properties and a turbulence dataset. We then conduct an examination of physical and biogeochemical properties
across frontal transects to identify and characterize subducting features. Finally, we calculate the turbulent erosion of a selected
intrusion of interest.
**2 Material and methods**
**2.1 Sampling strategy and profile inventory**
The study zone is highly dynamic and significantly influenced by the eastward-flowing Atlantic jet that sustains the WAG
(Sánchez-Garrido and Nadal, 2022). The jet is characterized by a pronounced frontal zone, exhibiting a density contrast of up
to 1.0 kg m$^{-3}$ at its boundaries (Oguz et al., 2014).
We conducted five transects across the salinity front identified through operational modeling and satellite estimations
in the northern edge of the WAG between March 28$^{th}$ and April 4$^{th}$ 2019 (**Fig. 1B**). Temperature and salinity conditions in the
upper water column were sampled with an Underway Conductivity Temperature Depth (UCTD) profiler, resulting in a total
of 136 profiles (mean depth: 231 m). Turbulence data were collected on 43 stations (mean depth: 219 m) during the campaign
using a microstructure profiler. With the exception of a single station, all stations featured duplicate microstructure profiles,
from which the mean value between these replicates was computed. Furthermore, we used a CTD probe to obtain 22 dissolved
oxygen and chlorophyll-a profiles (mean depth: 284 m) concurrently with the microstructure profiles.
**2.2. Temperature, salinity and derived variables**
Temperature (accuracy: 0.001 °C) and salinity (accuracy: 0.0003 S m$^{-1}$) data were acquired using a Teledyne RD Instruments
UCTD profiler, as detailed by Rudnick and Klinke (2007). The sampling rate is 16 Hz, with the UCTD falling velocity ranging
between 1.5 and 3.5 m s$^{-1}$. The spatial resolution between UCTD cycles was approximately 1 km, given a cruise speed of 3 m
s$^{-1}$ knots during recovery. The UCTD downcasts were post-processed for sensor alignment, salinity spikes correction and were
binned using a spline interpolation onto a vertical grid of 1 m. A comprehensive description of data post-processing procedures
can be found in Dever et al., (2019). Key oceanographic parameters, including Absolute Salinity, Conservative Temperature,
Brunt–Väisälä frequency (N$^2$), density ratio and Turner angle and spice were computed using the Gibbs Sea Water
oceanographic toolbox of TEOS-10 (https://www.teos-10.org/pubs/gsw).
N$^2$ serves as an indicator of water column vertical stability and was determined using equation (1):
$N^2 = -\frac{g}{\rho_0}\frac{\partial \rho}{\partial z}$                           (1)

where g represents gravitational acceleration (9.8 ms$^{-2}$), $\rho_0$ is a reference seawater density (1025 kg m$^{-3}$), and $\partial\rho/\partial z$ denotes the variability of potential density with depth. The density ratio quantifies the vertical contributions of Conservative Temperature and Absolute Salinity to the stability of the water column (following the Thermodynamic Equation of Seawater – 2010; IOC et al., 2010). The Turner angle, as outlined by McDougall et al., (1988), was computed to identify water column conditions, including double diffusivity (thermal diffusivity or salt fingering), stability, and instability regimes. Seawater spice, defined as the temperature and salinity variability along isopycnals, was employed to discern water masses with similar density, but varying temperature and salinity characteristics (McDougall et al., 2021). Spice anomaly was computed with respect to the mean spice profile computed in a temperature-salinity space (McDougall and Krzysik, 2015) and obtained including all spice profiles of the dataset. When computing the mean spice for a given density, we effectively combine specific temperature and salinity values that correspond to that density. The resulting spice anomaly then quantifies the deviation of the water parcel with the same density from the mean temperature-salinity combination, enabling the identification of intrusions. Furthermore, mixed layer depth was determined using a density threshold of 0.03 kg m$^{-3}$ relative to the reference density at 10 m depth, as proposed by de Boyer Montégut et al. (2004). Isopycnal strain, which measures the stretching or compression of isopycnal surfaces, was calculated as the vertical gradient of isopycnal displacement (Pinkel et al., 1991). This displacement is defined as the difference between the actual depth of each isopycnal and its expected depth based on the mean density profile calculated along the entire section.

**2.3. Detection of subducting intrusions**

Observational evidence of water being subducted from the upper ocean layer to below the mixed layer was observed by leveraging the high spatio-temporal resolution of the underway data collected by the UCTD. The presence of subsurface intrusions in a frontal transect was semi-automatically detected from the vertical profiles, based on subsurface spice and temperature anomalies. The detection algorithm proceeds as follows: I) Compute average spice on isopycnals for the campaign (auto). II) Compute spice anomaly on an isopycnal for each profile phase (auto). III) Detect subsurface anomalies in spice anomaly using a peak-finding algorithm based on peak prominence (auto). IV) Retain anomalies with at least 5 samples (i.e., 1.5 m; auto) and occur coherently over more than 3 consecutive profiles (manual).

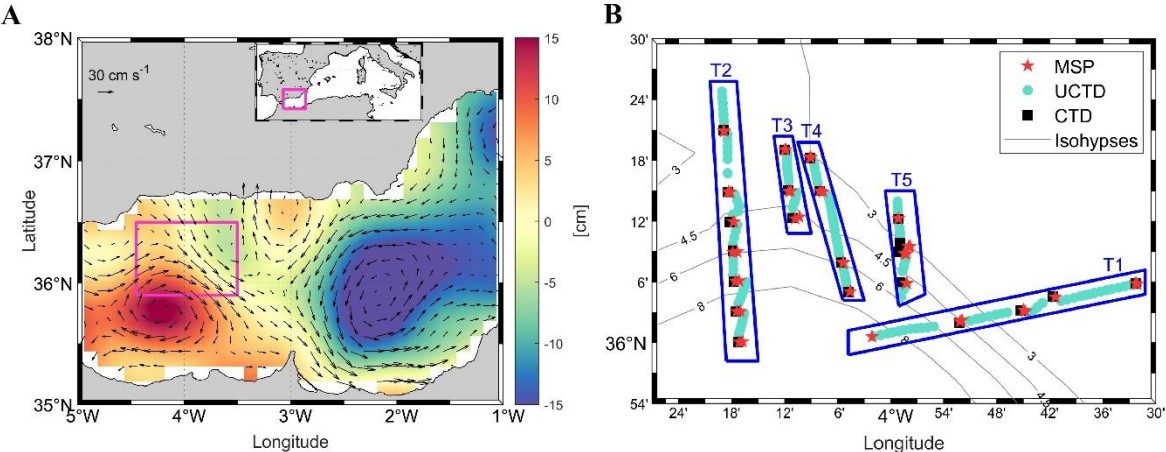

**Figure 1.** (A) Map of the Alboran Sea showing the mean absolute dynamic topography (colors) and geostrophic currents (arrows) on March 30th-31st, 2019. The purple inset shows the location of the sampling effort, detailed in panel (B), where blue rectangles denote the sampling stations selected for transect (T#) analysis. T1 was realized on March 29th, T2 on March 30th and T3-T5 on March 31st. Isohypses of absolute dynamic topography are depicted as gray lines. Red stars, cyan circles and black squares correspond to sampling stations for the microstructure profiler, underway CTD and CTD, respectively. Daily absolute dynamic topography and geostrophic current data were downloaded from https://data.marine.copernicus.eu/.

## 2.4. Dissolved oxygen and Chlorophyll-a

We equipped a SeaBird 911plus CTD probe with a SeaBird 43 dissolved oxygen sensor (accuracy: 2% of saturation) and a WET Labs ECO-AFL/FL fluorometer (sensitivity: 0.02 µg l$^{-1}$) to assess dissolved oxygen and chlorophyll-a concentrations, respectively. The CTD data underwent bin-averaging to achieve a vertical resolution of 0.5 m and was subsequently calibrated using in situ data. Dissolved oxygen estimates were aligned with measurements obtained through Winkler titration (n = 67; Mahadevan et al., 2020b), while chlorophyll-a estimates derived from fluorescence were calibrated against data from fluorometric determinations (n = 140; Alou-Font et al., 2019). A high level of agreement was found between in situ measurements and CTD-derived estimations, as evidenced by coefficient of determination ($R^2$) values of 0.99 for dissolved oxygen and 0.85 for chlorophyll-a.

Oxygen and chlorophyll-a anomaly on isopycnals were computed following equation (2):

$$X_a = X_\rho - \bar{X}_\rho \tag{2}$$

where $X_a$ represents the variable (i.e., oxygen or chlorophyll-a) anomaly, $X_\rho$ denotes the observed value at a specific density and $\bar{X}_\rho$ is the mean property value corresponding to this density.

## 2.5. Horizontal ocean currents, Richardson number and potential vorticity

Horizontal current magnitude and direction were collected using a hull-mounted Teledyne RDI Ocean Surveyor Acoustic Doppler Current Profiler (ADCP) operating at a frequency of 150 kHz and with a vertical bin size of 4 m. Detailed post-

processing procedures for ADCP data have been exhaustively documented in Mahadevan et al. (2020b) and Cutolo et al.
(2022). Shear squared ($S^2$) was calculated from ADCP data as the sum of the squares of the vertical gradients of the horizontal
velocity components (Gregg, 1989). This value was subsequently used to estimate the Richardson number for shear instabilities
(expressed as the ratio between $N^2$ and $S^2$; Cushman-Roisin and Beckers, 2011). Ertel potential vorticity (PV) was calculated
according to Zhmur et al. (2021) and equation (3):
$$PV = -(f + \zeta)\frac{\partial b}{\partial z} + \left(\frac{\partial v}{\partial z}\frac{\partial b}{\partial x} - \frac{\partial u}{\partial z}\frac{\partial b}{\partial y}\right) \tag{3}$$
where $f$ is the Coriolis parameter, $\zeta$ is the relative vorticity, $\partial b/\partial z$ corresponds to the vertical buoyancy gradient, $\partial b/\partial x$ and
$\partial b/\partial y$ are the horizontal buoyancy gradients, and $u$ and $v$ represent the horizontal current components.

## 2.6. Turbulent kinetic energy dissipation rates

Various methods have been employed to quantify turbulent mixing (e.g., integral approaches, finescale parameterizations and
direct microstructure measurements; Thorpe, 2005; Shroyer et al., 2018). In this study, we present turbulence dissipation rates
observations and derived parameters (Thorpe, 2005) collected using a free-falling microstructure profiler (MSS90D; Sea &
Sun Technology). The probe was equipped with two microstructure shear sensors (PNS6; sensitivity: $3.30 \cdot 10^{-4}$ and $3.82 \cdot 10^{-4}$
V m $s^2$ kg$^{-1}$ at 21 °C), with the final turbulent dissipation rate calculated as the mean of the two shear probe estimates. The
profiler's buoyancy was adjusted to achieve a sinking velocity between 0.6 and 0.7 m s$^{-1}$ and the data sampling occurred at a
frequency of 1024 Hz but was internally averaged to 512 Hz to comply with signal degradation along the 1.2 km probe cable.
Post-processing and turbulent dissipation rate calculations were carried out using the microstructure profiler processing toolbox
developed by Schulz et al. (2022). We fine-tuned instrument-specific parameters according to the microstructure profiler
employed in this study (e.g., sampling frequency, sensors calibration and sensitivity, distance of other sensors to the shear
sensor's tip), while the threshold parameters for data validation from Schulz et al. (2022) were retained. These processing
routines were evaluated using two benchmark ATOMIX (Analysing Ocean Turbulence Observations to Quantify Mixing)
datasets (Fer et al., 2024), which adhere to best practices for estimating dissipation rates from shear probes (Lueck et al., 2024).
The analysis showed strong consistency ($R^2 = 0.98$) between ATOMIX data processed using the Schulz et al. (2022) routines
and the Lueck et al. (2024) approach (**Supplementary Fig. 1**), with the former slightly overestimating dissipation rates by a
mean of 1.6%.
Kinetic energy dissipation rates ($\varepsilon$) were computed as per equation (4):
$$\varepsilon = \frac{15}{2}\nu\overline{\left(\frac{\partial u}{\partial z}\right)^2} \tag{4}$$
where $\nu$ represents the kinematic molecular viscosity and $\overline{(\partial u/\partial z)^2}$ is the spatial average of vertical shear variation with depth
(Taylor, 1935). Turbulent dissipation rates from both shear probes were treated separately, averaging all shear spectra within
1 m vertical bin. The shear spectrum results were iteratively fitted to the Nasmyth (Nasmyth, 1970) reference shear spectrum
and the deviation of the observed spectrum with respect to the Nasmyth's was used for data quality check. A detailed
description of the data processing procedure was described in Schulz et al. (2022). We performed data-averaging at 1-meter
depth intervals, excluding the initial 15 meters of each profile, to mitigate the noise arising from ship motion and wave-
breaking (D'Asaro, 2014).
The microstructure data exhibited good agreement ($R^2$ = 0.89) between the two shear sensors (n=8957;
**Supplementary Fig. 2A**), with a stronger correlation observed under elevated turbulence conditions ($\varepsilon > 10^{-7}$ W kg$^{-1}$) compared
to calmer waters ($\varepsilon < 10^{-7}$ W kg$^{-1}$). Another quality control parameter was the magnitude of the pseudo dissipation rates
originated from the profiler high frequency vibrations, consistently one order of magnitude lower than turbulent kinetic energy
dissipation rates (**Supplementary Fig. 2B**) and predominantly (36.9%) falling within the range of $1.0 \cdot 10^{-10}$ to $1.6 \cdot 10^{-10}$ W kg$^{-}$
$^{1}$.
**2.7 Turbulent fluxes**
Vertical diffusivity ($K_z$) is computed according to equation (5):
$K_z = \gamma \frac{\varepsilon}{N^2}$ (5)
where the mixing coefficient is $\gamma = 0.2$ (Gregg et al., 2018; Mouriño-Carballido et al., 2021; Lozovatsky et al., 2022), $\varepsilon$ is the
turbulent kinetic energy dissipation rate and $N^2$ denotes the squared buoyancy frequency.
We determine turbulent heat (in units of W m$^{-2}$) and salt fluxes (kg m$^{-2}$ s$^{-1}$) following Sheehan et al. (2023) and
equations (6) and (7):
$Q_H = -\rho_w C_p K_z \frac{\partial \theta}{\partial z}$ (6)
$Q_S = 10^{-3} \left( -\rho_w K_z \frac{\partial S}{\partial z} \right)$ (7)
where $\rho_w$ is seawater density, $C_p$ is the specific heat capacity of seawater (3850 J kg$^{-1}$ °C$^{-1}$), $\partial\theta/\partial z$ corresponds to the vertical
gradient of Conservative Temperature, and $\partial S/\partial z$ indicates the vertical gradient of Absolute Salinity. Furthermore, turbulent
fluxes of dissolved oxygen and chlorophyll-a (in units of mg m$^{-2}$ s$^{-1}$) were estimated using equations (8) proposed by Williams
et al. (2013):
$Q_X = -K_z \frac{\partial X}{\partial z}$ (8)
where $\partial X$ denote the variable (i.e., oxygen or chlorophyll-a) vertical gradient with depth.
Our analysis primarily focused on the subducting intrusion identified along transect 2 during the 2019 CALYPSO campaign
(**Fig. 1B**). The limited number of microstructure profiles precluded a comprehensive analysis of spatiotemporal intrusions
variability along the other transects. To assess the physical and biogeochemical conditions around the subducting intrusion
boundaries, we calculated the mean conditions within 5 m inside and outside the intrusion boundaries. The methodology used
to calculate diapycnal turbulent fluxes does not account for advective terms involving diapycnal velocity (Du et al., 2017) and
assume a constant mixing coefficient ($\gamma$) of 0.2. However, $\gamma$ can vary depending on stratification, turbulence intensity, and
water column regimes (Canuto et al., 2011; Gregg et al., 2018). Despite this, variability in $\gamma$ is typically smaller compared to
variations in turbulence (Le Boyer et al., 2023). It is important to note that the assumption of identical vertical diffusivity for
heat, salt, and tracers may introduce potential inaccuracies in flux estimates, which should be considered.

## 3 Results

### 3.1 Water column stability and turbulent kinetic energy dissipation rates

High mixing was observed in the surface layer, with localized turbulence peaks in the subsurface water column. Turbulent
kinetic energy (TKE) dissipation values displayed considerable variability, with 43.8% of observations falling between $1.3 \cdot 10^{-9}$
and $4.0 \cdot 10^{-9}$ W kg$^{-1}$ (mean ± standard deviation: $8.2 \cdot 10^{-9} \pm 2.4 \cdot 10^{-8}$ W kg$^{-1}$), with a peak (11.2%) identified in the range of
$2.0 \cdot 10^{-9}$ to $2.5 \cdot 10^{-9}$ W kg$^{-1}$. An analysis of ε probability distribution by depth intervals indicated that 95% of deep ε values were
comprised between $10^{-9}$ and $10^{-7}$ W kg$^{-1}$ (**Fig. 2A**). In contrast, surface and mid-water depths exhibited a lower proportion
(53% and 77%, respectively) within this ε range. Surface waters (<25 m) were characterized by elevated ε values, with 25%
and 12% of the data falling within the ranges of $10^{-7}$-$10^{-6}$ and $10^{-6}$-$10^{-5}$ W kg$^{-1}$, respectively.
Elevated homogeneity in the shallow water column vertical structure was observed. Indeed, the probability
distribution of Brunt–Väisälä frequency (N$^2$) by depth intervals (**Fig. 2B**) indicated lower stratification in the surface and mid-
water layers, where approximately 81% and 67% of values were lower than $0.2 \cdot 10^{-4}$ s$^{-2}$, respectively. Conversely, the deeper
portion (>50 m) of the water column exhibited stronger stratification, with an increased proportion (70%) of N$^2$ estimations
exceeding $0.3 \cdot 10^{-4}$ s$^{-2}$. These patterns were reflected in water column conditions. Examination of Turner angle values revealed
a predominantly stable water column, accounting for 74% of the dataset (**Fig. 2C**). However, these stability conditions
exhibited notable variations with depth. The shallow layer displayed a more varied scenario with a near-equal distribution
between statically unstable and doubly stable conditions. In contrast, the mid-water column featured the highest proportion
(27%) of diffusive regimes and the deep layer was primarily characterized (83%) by double stable conditions.

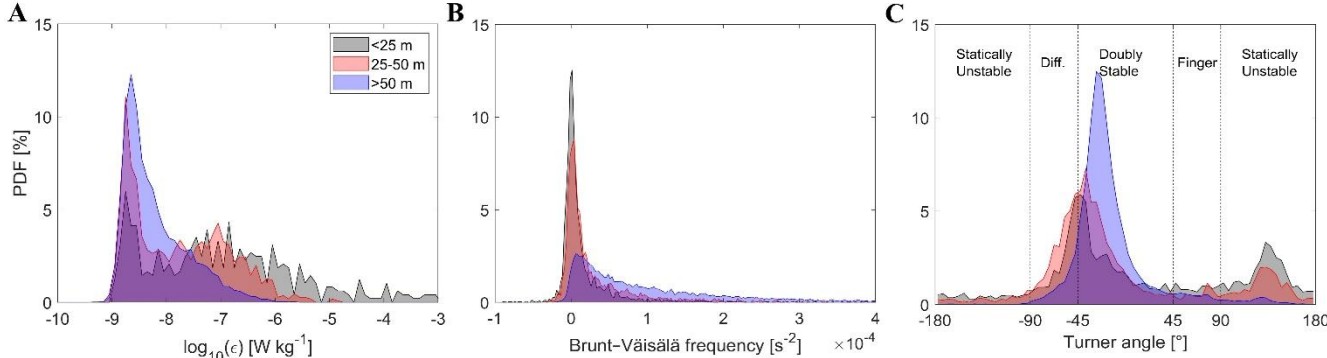


**Figure 2.** Probability distribution frequency (PDF) by depth intervals for turbulent kinetic energy dissipation rates (A), Brunt–Väisälä frequency (B) and Turner angle values (C). Colored shaded areas in panels correspond to different depth intervals, with gray: 15-25 m; red: 26-50 m; blue: depths >51 m. The names in panel (C) reflect the water column regime according to the Turner angle value (McDougall et al., 1988).

231

## 3.2 Transects across the Western Alboran Gyre front

A noticeable depression in the isopycnals was consistently observed in all the transects extending towards the interior of the anticyclone (**Fig. 3** and **Supplementary Fig. 3-7**). The highest ε below the mixed layer was detected adjacent to zones featuring elevated vertical density gradients and deepening along the isopycnals in transects 2 (**Fig. 3**). A deepening of positive spice anomalies from approximately 50 to 100 m was observed at the start of transect 1 and from 15-35 km of transect 2. Subducting intrusions were observed along all transects, except for transect 3, possibly owing to its shorter length (approximately 12 km; **Fig. 3** and **Supplementary Fig. 3-7**). The mean thickness of subducting intrusions was computed at 14.2 m (standard deviation: 9.4 m), ranging from a minimum of 1.7 to a maximum of 42.2 m. The subduction is likely occurring along the frontal direction, following the anticyclonic circulation, rather than necessarily along the tilted isopycnals identified in the transect.

Enhanced ε and diffusivity values were noted in proximity to the base of the mixed layer and in the vicinity of subducting intrusion boundaries (**Fig. 4**). Furthermore, diffusive water column conditions were identified along the upper boundary of the subducting intrusion in transect 2 (**Fig. 3D**) and adjacent to the subducting intrusions within transects 1 and 4. Positive isopycnal strain values were observed at both edges of the subducting intrusion initially, with a predominant concentration of positive values indicating stretching of isopycnal surfaces primarily at the bottom edge as subduction progressed (**Fig. 3E**). The current data along transect 2 illustrated a horizontal velocity magnitude exceeding 60 cm s$^{-1}$ within the interior of the anticyclone, while lower values were observed on its periphery. The subducting intrusion, identified beneath the superficial high-velocity patch and within a zone of elevated shear squared (primarily due to a negative vertical gradient of the zonal velocity component; **Fig. 3F**), was characterized by a horizontal velocity estimated at approximately 0.5 m s$^{-1}$. The mean Richardson number across the transect was calculated to be 0.89, indicating a generally stable water column with respect to share instabilities. However, lower Richardson numbers were observed in the initial zone of the intrusion (between km 17 and 29 of the transect; **Fig. 3G**). No significant correlation was found between shear and ε, suggesting that stratification may suppress shear-driven turbulence and/or that other sources of turbulence could be influencing the study area. The subducted water exhibited positive potential vorticity (**Fig. 3H**), revealing unfavourable conditions for the generation of symmetric instability.

A deepening of the well oxygenated surface layer towards the center of the anticyclone was observed in transects 1 and 2 (**Supplementary Fig. 3** and **Fig. 3**, respectively). Elevated dissolved oxygen anomaly concentrations (>0.5 mg l$^{-1}$) were detected inside the subducting intrusion along transect 2 (**Fig. 3I**), with high values deepening from approximately 50 to 120 m. Similarly, anomalous high chlorophyll-a anomaly values were found near the 30 km of transect 2, with anomaly concentrations of up to 1.4 mg m$^{-3}$ detected at a depth of 100 m (**Fig. 3J**).

261

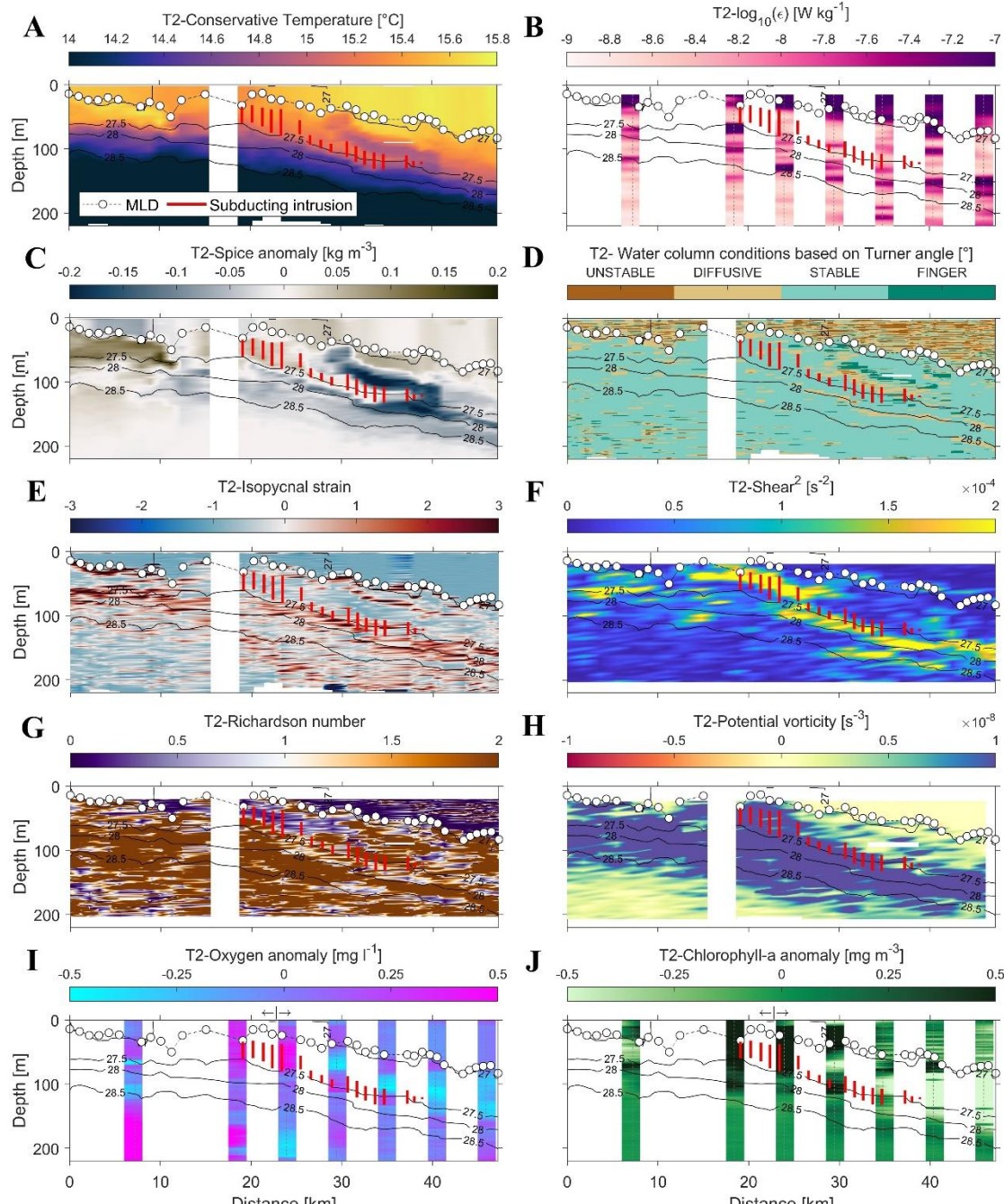

262

**Figure 3.** Profiles of Conservative Temperature (A), turbulent kinetic energy dissipation rates (B), spice anomaly (C), water column conditions based on Turner angle estimations (D), isopycnal strain (E), shear squared (F), Richardson number (G), potential vorticity (H), dissolved oxygen anomaly (I), and chlorophyll-a anomaly (J) estimations acquired along transect 2 of the 2019 CALYSPSO campaign. Isopycnals are represented as black lines, while the mixed layer depth and subducting intrusions are denoted by colored points and lines, respectively. The distances between stations were calculated starting from the northernmost sampling point.

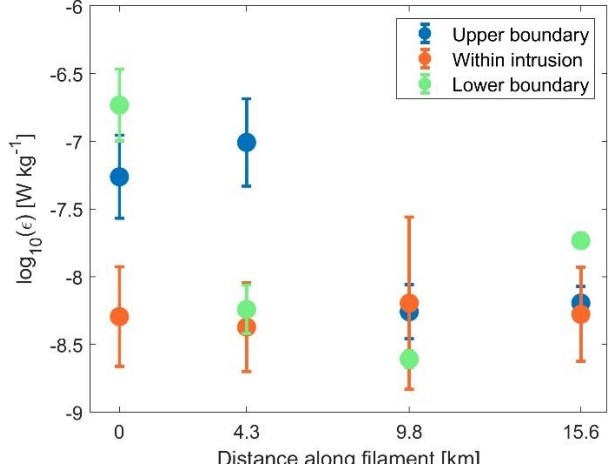

**Figure 4.** Mean turbulent kinetic energy dissipation values within the intrusion and at its upper and lower boundaries (5 m from the intrusion edges). The errorbars represent the measurement standard deviation.

### 3.3 Turbulent fluxes around the subducting intrusion

We focused our analysis of the microstructure profiles to transect 2 during the 2019 CALYPSO campaign due to its higher horizontal resolution (**Fig. 1B**). Turbulent fluxes within the interior of the transect 2 intrusion exhibited reduced values compared to water column around both intrusion edges (**Supplementary Fig. 8**). Notably, turbulent fluxes exhibited higher magnitudes within the first two profiles sampling the edges of the subducting intrusion in comparison to the subsequent two profiles (**Fig. 5**). Turbulent fluxes around the intrusion boundaries resulted in a net loss of heat, oxygen and chlorophyll-a properties from within the intrusion to the surrounding ocean, while salinity increased (**Table 1**). Heat, oxygen and chlorophyll-a turbulent fluxes revealed a consistent properties loss at the base of the intrusion towards the deeper layer, while salt fluxes displayed a coherent property loss (gain) at the upper (lower) boundary of the intrusion. Heat loss was consistently recorded near the upper intrusion boundary at all sampling stations, although more variability was observed in oxygen and chlorophyll-a fluxes. The mean absolute values for turbulent fluxes indicated reduced heat, oxygen and chlorophyll-a fluxes near the upper boundary in contrast to the intrusion's base. Specifically, the upper heat flux accounted for only 35% of the magnitude observed near the base of the intrusion, while the upper oxygen and chlorophyll-a fluxes represented 68 and 63%, respectively, of the corresponding bottom flux magnitudes. The fluxes uncertainty was provided in **Supplementary Table 1**.

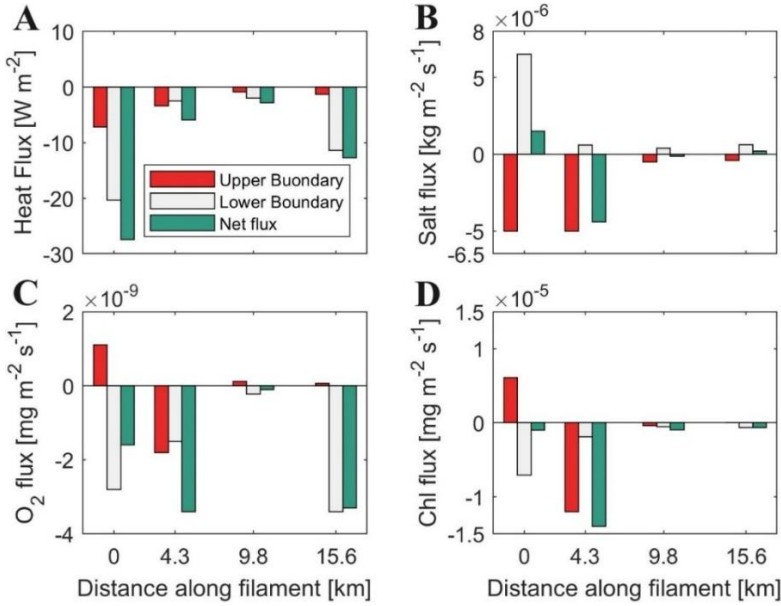


**Figure 5.** Estimations of turbulent fluxes of heat (A), salt (B), oxygen (C) and chlorophyll-a (D) along the upper and lower boundaries of
the subducting intrusion identified within transect T2 of the 2019 CALYPSO campaign and the resulting net flux (in green). The distances
between the four stations where the fluxes were calculated (as shown in Supplementary Figure 8) along the subducting intrusion are provided.
Positive (negative) values for the turbulent fluxes represent a gain (loss) of the respective variables within the interior of the intrusion.

**Table 1.** Mean (± 95% confidence interval) Conservative Temperature, Absolute Salinity, dissolved oxygen and chlorophyll-a conditions
within the subducting intrusion identified along transect T2 and estimations of daily turbulent heat, salt, oxygen and chlorophyll-a fluxes.
The fluxes were computed as the rate of change of properties [(delta flux) (intrusion width)$^{-1}$]. Negative (positive) values denote a loss (gain)
within the interior of the intrusion. The distances between the four stations, the intrusion mean depth and thickness are provided.

| Variable | Distance along transect [km] | | | |
|---|---|---|---|---|
| | **0** | **4.3** | **9.8** | **15.6** |
| **Intrusion characteristics** | | | | |
| Depth [m] | 47 | 59 | 99 | 120 |
| Thickness [m] | 25.9 | 42.2 | 13.3 | 24.9 |
| **Mean properties** | | | | |
| Temperature [°C] | 15.28 ± 0.05 | 15.27 ± 0.06 | 15.16 ± 0.03 | 14.91 ± 0.06 |
| Salinity [g kg$^{-1}$] | 37.03 ± 0.03 | 37.04 ± 0.04 | 37.10 ± 0.03 | 37.15 ± 0.07 |
| Oxygen [mg l$^{-1}$] | 7.74 ± 0.07 | 7.84 ± 0.10 | 7.43 ± 0.05 | 7.16 ± 0.04 |
| Chlorophyll-a [mg m$^{-3}$] | 1.81 ± 0.12 | 2.16 ± 0.28 | 1.38 ± 0.13 | 0.68 ± 0.02 |
| **Daily fluxes** | | | | |
| Heat [°C d$^{-1}$] | $-2.2 \cdot 10^{-2}$ | $-2.9 \cdot 10^{-3}$ | $-4.4 \cdot 10^{-3}$ | $-1.1 \cdot 10^{-2}$ |
| Salt [g kg$^{-1}$ d$^{-1}$] | $5.1 \cdot 10^{-3}$ | $-9.1 \cdot 10^{-3}$ | $-7.6 \cdot 10^{-4}$ | $7.0 \cdot 10^{-4}$ |
| Oxygen [mg l$^{-1}$ d$^{-1}$] | $-5.5 \cdot 10^{-3}$ | $-6.9 \cdot 10^{-3}$ | $-6.7 \cdot 10^{-4}$ | $-1.1 \cdot 10^{-2}$ |
| Chlorophyll-a [mg m$^{-3}$ d$^{-1}$] | $-3.4 \cdot 10^{-3}$ | $-2.9 \cdot 10^{-2}$ | $-6.2 \cdot 10^{-3}$ | $-2.2 \cdot 10^{-3}$ |

## 4 Discussion

### 4.1 Turbulent kinetic energy dissipation rates in the Western Alboran Sea

The TKE dissipation rates in our study were (mean: $8.2 \cdot 10^{-9}$ W kg$^{-1}$) found to be comparable to those reported in previous
investigations involving microstructure data in the Mediterranean Sea. For instance, Cuypers et al. (2012) calculated mean
TKE dissipation values of approximately $10^{-8}$ W kg$^{-1}$ below the seasonal pycnocline. Our TKE dissipation estimates unveiled
an intermediate turbulent environment, between the Mediterranean energetic and quiescent regions (mean: $5.2 \cdot 10^{-8}$ and $4.7 \cdot 10^{-10}$
$^{10}$ W kg$^{-1}$, respectively; Vladoiu et al., 2021). Interestingly, our findings exhibited a closer resemblance to the TKE dissipation
observed west of the Gibraltar Strait, where the mean TKE dissipation was $4 \cdot 10^{-9}$ W kg$^{-1}$ in the ocean interior (Fernández-
Castro et al., 2014).
The observed peaks in TKE dissipation rates were predictably situated in shallow ocean regions influenced by wave
breaking, in close proximity to the base of mixed layer (Zippel et al., 2022) and near the boundaries of subducting intrusions
(**Fig. 3**). However, other peaks were detected at deeper levels and did not appear to correlate with aforementioned factors.
Mixing processes in the stratified ocean below the mixed layer are often attributed to vertical shear extending below the MLD,
penetrative convection and the breaking of internal waves (MacKinnon et al., 2013). The Western Alboran Sea may be
influenced by the eastward propagation of internal waves traveling along isopycnals generated by the interaction of tidal
currents with bathymetry at the Gibraltar Strait (Thorpe, 2007; Alpers et al., 2008; Bolado-Penagos et al., 2023). While
symmetric instabilities have been identified as effective mechanisms for geostrophic energy dissipation in the ocean interior
(Zhou et al., 2022), the positive sign of the potential vorticity associated with subducting water in the study area (**Fig. 3H**)
suggests that the conditions required for this process to occur may not be met. Another plausible explanation for the deep TKE
dissipation peaks could be provided by dissipation associated with subducting intrusions that may have gone undetected by
our methodology. Conducting future surveys with mooring and/or glider deployments to identify internal waves within the
study zone could significantly advance our comprehension of their spatiotemporal variability and their role in generating deep
turbulence along isopycnals.

### 4.2 Water column regimes

The convergence of Atlantic and Mediterranean waters in the study zone resulted in a robust stratification of the water column,
characterized predominantly by doubly stable conditions. Along isopycnals and at the upper boundary of the subducting
intrusion (**Figure 3**), we observed instances of diffusive convection regimes. While diffusive convection is typically associated
with thermohaline staircases and is more commonly found at higher latitudes (Kelley et al., 2002; van der Boog et al., 2021),
the presence of horizontal variability in temperature and salinity conditions in our study area may lead to the formation of
coherent subducting intrusions associated with double diffusive convection (Kelley et al., 2002; Schmitt, 2009). Freilich and
Mahadevan (2021) proposed that the specific pathway of subducting intrusions along isopycnals in the study zone could be
generated by a combination of mesoscale (geostrophic) frontogenesis and submesoscale (ageostrophic) dynamics.
The subducting intrusion transports subsurface water column properties into the deeper ocean, undergoing erosion
along its pathway through a combination of turbulent and diffusive mixing. This dynamic process results in a modification of
its inherent properties.

**4.3 Turbulent erosion of the intrusion**

The elevated TKE dissipation rates in the surface layer, coupled with an increase in stratification with depth can potentially
account for the higher diffusivity and turbulent fluxes observed at the start of the intrusion's subduction compared to stations
sampled further along the subduction path. Moreover, physical and biogeochemical properties of the subducted water
resembled surface conditions more closely than those of the deep layer, resulting in reduced fluxes along the upper boundary
of the intrusion compared to the lower boundary (with the exception of the station located at 4.3 km).
The turbulent erosion of the subducting filament led to an overall decrease in temperature, oxygen and chlorophyll-a
content within the filament, while salinity increased (**Table 1**). The slight increment in oxygen and chlorophyll-a concentration
observed at the second station within the intrusion may be attributed to either the properties gain detected at the upper boundary
of the first station, indicating a supply of biogeochemical properties from the surface layer into the intrusion interior, or in situ
phytoplankton production (the photic layer was estimated to be around 60 m deep; **Supplementary Fig. 9**).
However, these diapycnal fluxes were too weak to induce a significant dilution of the intrusion, as daily fluxes were
orders of magnitude smaller than the mean property values within the intrusion (**Table 1**). These estimates did not account for
double-diffusive mixing fluxes characteristic of thermohaline staircases, as such features are predominant at greater depths in
the Western Mediterranean Sea (Onken and Brambilla, 2003; Schroeder et al., 2016; Ferron et al., 2021). Despite of this, the
estimates of diffusive convection mixing were negligible compared to the turbulent fluxes. In addition to turbulent and
diffusive convection mixing, the water column density stability and isopycnal mixing might contribute to the typical vertical
variability in subsurface ocean temperature and salinity. Specifically, isopycnal mixing might act an important role in the
observed dilution given its contribution in meso- and submesoscale coherent features (Abernathey et al., 2022). Conversely,
the decline in oxygen and chlorophyll-a content with depth can be attributed to the deepening of the photic layer, distance from
the atmospheric-ocean boundary layer, and processes such as remineralization, respiration, and grazing. The modification of
the typical vertical variability in biogeochemical properties induced by subducting intrusions might have profound impacts on
ecosystem dynamics within the study zone.

**4.4 Biogeochemical significance of subducting intrusions**

The Atlantic Jet, which enters the Mediterranean through the Strait of Gibraltar, coupled with coastal upwelling events,
transforms our study area into one of the most productive zones in the Mediterranean despite the Mediterranean Sea's well-

known status as an oligotrophic basin (Reale et al., 2020; Sánchez-Garrido and Nadal, 2022). The outer boundary of the WAG has also been identified as a stirring region where properties of the water column are continually exchanged as they are advected towards the center of the anticyclone (Brett et al., 2020; Sala et al., 2022). Subduction of the intrusion may enhance particulate organic carbon export below the mixed layer, reducing its exposure time to remineralization (Freilich et al., 2024). This process contributes to one of the highest export rates observed in the Mediterranean Sea based on sediment trap and particle size distribution profiles data (Ramondenc et al., 2016). Additionally, the mixing associated with subducting intrusions may facilitate the reorganization of phytoplankton communities, traditionally stratified in the photic layer (Mena et al., 2019) and their proliferation. This is especially significant, as nitrates are nearly depleted in the shallow layer north of the WAG (Oguz et al., 2014; Lazzari et al., 2016; García-Martínez et al., 2018). It has been demonstrated that oceanic fronts might act as aggregation areas for planktonic organisms, becoming important foraging areas for higher trophic layers (Acha et al., 2015). Moreover, the transport of chlorophyll-a towards the center of the WAG could lead to an increase in the biomass of diel vertical migrant zooplankton, which tends to be more abundant in the inner part of the gyre compared to its periphery (Yebra et al., 2018).

## 5 Conclusions

The Western Alboran Gyre is a dynamical feature characterized by high spatiotemporal variability arising from the convergence of Mediterranean and Atlantic waters. Indeed, the northern edge of the WAG water column exhibited notable spatial variability in both physical and biogeochemical characteristics. Specifically, the inner part of this gyre featured higher temperature, current velocity, oxygen content and chlorophyll-a concentration compared to its periphery. Moreover, there was an observable deepening of enhanced Brunt–Väisälä frequency and turbulent kinetic energy dissipation rates towards the anticyclone's center.

The investigation of spice anomaly spatial variability allowed the identification of several subducting intrusions occurring beneath the mixed layer depth, extending from the gyre's outer region towards its center. High turbulent kinetic energy dissipation rates were evident at both the upper and lower boundaries of these intrusions, complemented by localized peaks at deeper levels. The specific factors contributing to these heightened dissipation rates at deeper levels remain elusive.

The turbulent fluxes of heat, salt, oxygen and chlorophyll-a along the intrusion boundaries revealed a consistent net loss of physical and biogeochemical properties from within the intrusion to the surrounding ocean. From a biogeochemical perspective, the subduction intrusion holds significance as it has the potential to amplify the export of particulate organic carbon below the mixed layer. Additionally, it may contribute to the enhancement of diel vertical migrant zooplankton biomass and facilitate the proliferation of phytoplankton communities. Notably, mixing due to turbulence or diffusive convection contributed little to the observed variation in temperature, salinity, oxygen or chlorophyll-a within the intrusion interior. Other factors, such as water column density stability, variability of the photic layer depth, and organic matter degradation, likely played a role in these dynamics.

While our present study has provided valuable insights into the subduction of intrusions and their turbulent erosion
within the Western Alboran Gyre, significant gaps remain in our understanding of the spatiotemporal variability of subducting
intrusions. Future targeted surveys that specifically address the persistence and extent of these features might improve
quantitative parametrizations of key physical and biogeochemical property subduction. Explorations encompassing a broader
surface of the WAG may reveal asymmetries in intrusion subduction between the WAG's edges and offer an estimate of the
total subduction occurring within the WAG.

*Funding.* CALYPSO constitutes a Departmental Research Initiative funded by the U.S. Office of Naval Research. GT was
founded by ISMAR-26-2022-VE and ISMAR-18-2023-VE research fellowships. TMSJ was supported by ONR grant N00014-
400  18-1-2416.


*Data availability.* The dataset used in this study is stored and openly shared via Zenodo (Testa et al., 2025), a multidisciplinary
repository maintained by CERN (European Organization for Nuclear Research).

*Author contributions.* GT: Conceptualization, Methodology, Software, Validation, Formal analysis, Data Curation, Writing -
Original Draft, Writing - Review & Editing, Visualization. MD: Methodology, Software, Data Curation, Writing - Original
Draft, Writing - Review & Editing, Supervision. MF: Resources, Writing - Review & Editing. AM: Resources, Writing -
Review & Editing, Project administration, Funding acquisition. SJ: Resources, Writing - Review & Editing. LP: Resources,
Writing - Review & Editing. FF: Conceptualization, Methodology, Software, Data Curation, Writing - Original Draft, Writing
- Review & Editing, Supervision, Project administration, Funding acquisition.

*Competing interests.* The authors declare that they have no conflict of interest.

*Acknowledgements.* We extend our sincere appreciation to the captains and crews of the R/V *Pourquois Pas?*, as well as the
technical and scientific personnel involved in making measurements and providing support. The authors wish to express their
gratitude to Leo Middleton engaging in insightful conversations that influenced the development of this article. Furthermore,
we would like to acknowledge all the CALYPSO researchers whose constructive comments during CALYPSO's Padua
meeting enriched this study. Finally, we thank the editor and reviewers for their valuable assistance, comments, and
suggestions.

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
