# Peer review of "Turbulent erosion of a subducting intrusion in the Western"

_EGUsphere, 2024_

## Author Response (AR1)

We sincerely thank the editor and reviewers for their insightful comments and observations on the initial manuscript. We believe their suggestions have significantly enhanced the methodology, results presentation, and discussion in this revised version. All issues raised were carefully addressed, and we hope this version meets the reviewers' expectations and demonstrates substantial improvement.

Key changes in the revised manuscript include:
- The calculation of additional parameters (Rossby number, potential vorticity, and Richardson number for shear instabilities) to better describe the dynamic conditions of the study area.
- A comparison of the processing routines with the ATOMIX datasets.
- The removal of double diffusion estimates.
- The inclusion of a discussion on the methodology's limitations.
- Revisions to improve clarity and readability throughout the manuscript.

Below, we provide detailed responses to each of the editor's and reviewers' comments:

**EDITOR**

**Comment:** I kindly request that you consider testing your microstructure data processing method against several benchmark datasets produced under the SCOR working group ATOMIX. You can find more information on the shear probes group wiki: https://atomix.app.uib.no/Shear_probes. I was involved in the Schulz et al. (2022) study, which was conducted prior to the finalization of the ATOMIX recommendations. There are several points where the IOW routines (the version used in that study) may differ from the ATOMIX best practices described in Lueck et al. (2024): https://www.frontiersin.org/articles/10.3389/fmars.2024.1334327/full. It would be highly beneficial for both the readers and reviewers if you could test your routines against one or several of the five benchmark ATOMIX datasets described in Fer et al. (2024): https://www.nature.com/articles/s41597-024-03323-y. These datasets are freely accessible from the wiki and through the links provided in Fer et al., via the British Oceanographic Data Centre. You might consider including some comparison plots in your response letter or as supplementary material. However, I strongly encourage you to include a statement in your manuscript regarding the results of this comparison. If you observe significant and systematic deviations from the ATOMIX dissipation rate estimates, reprocessing may be necessary. The benchmark data are provided from full-resolution profiles (level 1) of shear probe data through spectra to dissipation profiles (level 4). Therefore, passing the time series of level 1 (or the cleaned data in level 2) through your routines and comparing them to the dissipation profiles in level 4 should be a straightforward task.

**Response:** We sincerely appreciate the editor time and effort in reviewing our manuscript. We are also grateful for bringing the recent ATOMIX benchmark datasets to our attention, as we were previously unaware of them. We believe agree that sharing these datasets represents an important step in analyzing future turbulent kinetic energy dissipation rates derived from shear probe measurements.
As suggested, we applied the routines described by Schultz et al. (2022) to Level 2 data from two ATOMIX profiles: one collected in the Baltic Sea using an MSS90L profiler and the other in the Haro Strait using a VMP250 instrument. A statement summarizing the results of this comparison has been added to the main text, along with a figure included as supplementary material in the revised manuscript.
The comparison demonstrates good coherence between the two procedures, with no evident or systematic differences observed in the analyzed ATOMIX profiles. We noted that a higher percentage of data was retained when using the Schultz et al. (2022) procedure, likely due to differences in the quality-assurance metrics employed by the two methods. In future studies, we will ensure that data processing adheres to the best practices outlined in Lueck et al. (2024).

**REVIEWER #1**

This work discusses the turbulent dissipation rates and corresponding tracer fluxes associated with an interleaving feature observed within the Western Alboran Gyre. They used microstructure data and biochemical measurements to characterize the interleaving feature and describe how it can potentially erode within the water column. Using other parameters such as stratification and turner angle, they classify the water column and discuss possible mechanisms driving dissipation. The manuscript is well written and can contribute to further understanding the relationship between interleavings and dissipation and provide insight into the fluxes within the WAG. After further clarifying parts of the data analysis, this paper can be a good addition to the literature. Please see my comments below.

Specific comments:
**Comment (C)**: In the methods sections, It was hard to follow what type of instruments were used in section 2.1. What are the profiles mentioned here? Are these UCTD profiles? I believe so after reading further, but I suggest that more information is provided about the instruments before quantifying profiles.

**Response (R):** We agree with the reviewer. We have specified the instruments used to collect the data in this section to help readers understand how each variable was measured.

**C:** Further discussion about the assumption that all diffusivity values (Tt, Ks, Ko) are the same is needed. Even more, as the paper mentions, are double diffusion instabilities. What are the limitations of estimating epsilon in a DD regime and assuming a .2 gamma? Is there a relationship between elevated epsilon and shear in the data set? This last point might have been mentioned, but I could not find it; I suggest highlighting it and making it clearer. Even though Tu indicates DD, it might not be present in the area.

**R:** This is a valuable comment, and we have addressed it in the revised manuscript. We decided to exclude the double diffusion fluxes; the reasons for this decision are explained in a subsequent response. We have also included a discussion of the limitations of the methodology, as suggested. Regarding the potential correlation between shear and epsilon, our analysis found no correlation between the two variables. Possible explanations for this lack of correlation are now included in the revised manuscript.

**C:** Lines 231-235: There seems to be an elevated Oxygen patch at 6 km, below 100 m depth. Can you elaborate a bit more about this subsurface Oxygen maximum?

**R:** We thank the reviewer for the useful comment. Indeed, Figure 3I suggests a subsurface oxygen peak at the mentioned location. However, please consider that the panel shows oxygen anomalies data, calculated as the deviation from the mean value on isopycnals. We believe that the positive anomaly represents higher oxygen concentrations (a mean value of 6.1 mg l$^{-1}$ was calculated between 120 and 220 m) observed at this location compared to the other part of the transect (e.g., 5.8 mg l$^{-1}$ at 18 km and 6.0 mg l$^{-1}$ at 29 km) within the same density range. Furthermore, in Supplementary Figure 3H, which shows the absolute dissolved oxygen concentration, no subsurface oxygen maximum is observed.

**C:** Line 257-259: It was hard to follow what positive fluxes the authors are referring to here. From Figure 5, it appears that, for example, Heat flux is negative in both boundaries.

**R:** The reviewer is right. We have rephrased this sentence, removing the positive/negative terminology, which referred to upward/downward flux directions. We believe this classification was confusing, so we now only mention the gain or loss of properties, in line with Figure 5.

**C:** Line 324-327: The authors include an estimate of double diffusion dissipation here, but the explanation of how they did this is in the supplement material and very briefly described. The equations they use from Nagai et al. and Nakano et al are parameterizations using fine-scale values of Density ratio, but they did not discuss the limitations of these parameterizations, if the coefficients of this equation are valid in the Mediterranean Sea, or discuss other parameterizations (as mentioned in Nakano et al.). I think it is important to discuss the implication of the differences in fluxes between turbulent dissipation and DD, but if Double diffusion is included in the paper, I think there has to be more information in the paper about it, not only in the supplement material.

**R:** We appreciate the reviewer's insights on this point and we agree with it. After internal discussion with the co-authors we decided to eliminate the double diffusion estimates, as their magnitude was negligible. This analysis was considered unnecessary for the paper's main message and could potentially confuse readers.

Technical corrections

**C:** Line 100: It might be helpful to clarify spice estimated from the TEOS-10 is known as "spiciness."

**R:** Done.

**C:** Line 192 and Figure 4: Is it supposed to be TKE dissipation? Not just TKE? Throughout the manuscript, there are several places where the word dissipation is missing, and only TKE is stated.

**R:** The reviewer is correct. We carefully reviewed the manuscript and added the word "dissipation" where needed.

**C:** Line 202: I don't understand what " water column regimes" is referring to here

**R:** The term "regimes" has been replaced with "conditions".

**C:** Line 221: I'm unsure what the starting sentence refers to. Can you elaborate?

**R:** This sentence has been rephrased for better clarity.

**C:** Figure 7 supplement: It is hard to distinguish between the colors of Chlorofphul fluxes in this figure, when they are positive and when they are negative. The color bar is saturated.

**R:** We have updated the colorbar for chlorophyll-a turbulent fluxes in Supplementary Figure 7 in the revised manuscript, following the reviewer suggestion.

**REVIEWER #2**

The paper by Testa et al presents an interesting data set of a subducting intrusion in the western
mediterranean sea. Subducting intrusion play an important role in ocean ventilation and export of
biogeochemical properties. The observations are interesting because they are based on high
resolution hydrographic profiles and also present some turbulence profiles which allow
quantification of turbulent fluxes of properties.

The paper will therefore be a valuable contribution afer some moderate revision regarding a
clarification of the dynamical context and of some of the methods (I chose major, as revisions may
be important for publication but they should be adressed quite easily I think)-

**Comment (C):** The dynamical context needs further clarification,  to cite the author "However, so
far there has been limited research that specifically identifies occurrences of quasi-balanced
subsurface vertical velocity and examines how turbulence responds to such instances."  This
appears to be a motivation for the study, but the analysis is quite elusive regarding the dynamical
conditions prevailing in the eddy (Rossby number, potential vorticity and symmetric instability
potential, Richardson number for shear instabilities) these parameters can likely be estimated using
the observations.

**Response (R):** The reviewer raises a very important point. To provide a more comprehensive
analysis of the dynamical conditions in the eddy, all the suggested parameters were calculated and
incorporated into the revised manuscript. We believe these additions enhance the discussion of the
eddy's dynamical context.

**C:** The definition of the subducting water is not very clear, I am not familiar with the spice concept,
from your definition it is the temperature and salinity variability along isopycnals… so it mixes
temperature and salinity, is it a normalized variability? can you provide an explicit expression?
Spice anomaly would then be an anomaly relative to a mean variability of salinity temperature
along isopycnals, it is hard to me to relate this with subducting intrusion can you explain the
relationship.

**R:** The reviewer's observation is appreciated, as the spice anomaly concept might indeed be unclear
to non-specialists. To address this, a reference to McDougall and Krzysik (2015) has been added to
the manuscript, where interested readers can find further information. Additionally, the explanation
was expanded to clarify why the anomaly was calculated and how it was utilized to identify
subduction intrusions, making it more accessible to a wider audience.

I have a few interrogations regarding fluxes and impact on dilution (see detailed comments)

**C:** I think it would be interesting to have panels of salinity and velocity in Fig.3 like in the
supplementary material. In general why did you put so much stuff in supplementary instead of the
main text (space restriction?)

**R:** This is a valid observation. Salinity and velocity panels were not included in Fig. 3 because the
intrusion signal was not particularly evident in the salinity distribution for this section. While
horizontal velocity patterns, such as the surface high-velocity patch in the cyclone, are interesting
(and mentioned in the main text), we prioritized displaying shear squared, as it provides more direct
insight into intrusion location due to its calculation as the vertical gradient of horizontal velocity
components. The decision to include substantial material in the supplementary section was driven
by space restrictions and an effort to focus the main text on figures most relevant to the study's core message. In response to the other reviewer feedback, the supplementary material has been
streamlined and refined.

Specific comments

**C:** L46-50 do you mean three dimensional turbulence here (~ mixing)

**R:** Yes, this is correct. The sentence was revised to clarify this point and improve its readability.

**C:** L104-106 Isopycnal strain, how is computed the mean density profile? Is it the mean density
profile over the full section?

**R:** The reviewer's interpretation is correct; the mean density profile was calculated over the entire
section. This clarification has been added to the revised manuscript.

**C:** Figure 5 I am quite confused with the computation of the net fluxes. Do you try to get the net
flux going into the intrusion, if so I can't see a situation where two fluxes of the same sign could
add up (one is necessarily exiting the layer) moreover it seems to me that computing a rate of
change of properties  (Delat of Flux/ intrusion width) would be more interesting as it could be more
directly related to a time scale for the dilution of the intrusion properties

**R:** The sentence at lines 257–259 of the original manuscript has been rephrased to remove the
potentially confusing positive/negative terminology related to flux direction. To clarify, the flux
signs in Fig. 5 represent: positive = net property gain inside the intrusion; negative = net property
loss from the intrusion to the exterior. For example, in the first station of the heat fluxes, negative
values at both the upper and lower boundaries indicate heat loss at both edges. The resulting net
flux represents the total loss, calculated as the sum of the two edge fluxes. The rate of change of
properties is indeed an interesting metric and has been calculated as daily fluxes, as described in the
response below.

**C:** L320 321 "However, these diapycnal fluxes were too weak to induce a significant dilution of the
intrusion, as daily fluxes (**Supplementary Table 2**) were orders of magnitude smaller than the
mean property values within the intrusion". It is difficult to compare fluxes and mean properties it
has different units. Here again it seems to me that you should compute a flux divergence that will
give you a rate of change per unit volume and then you can divide a variation of property along the
intrusion to get a typical time scale of dilution of this property by turbulent fluxes

**R:** The reviewer is right. Indeed, the fluxes shown in Supplementary Table 2 in the original version
of the manuscript were computed as the daily rate of change of properties (delta flux/intrusion
width). As such, these flux estimates share the same units as the mean property values calculated
inside the intrusions (Table 1). A more detailed explanation of how these fluxes were calculated has
been added to the caption of Table 1. This update reflect the inclusion of daily turbulent fluxes in
Table 1, which followed the decision to exclude double diffusivity fluxes.

**C:** L329-330 I am not sure that solar radiation and evaporation transpiration have a significant
impact below 50 m, not on salinity for instance, or if it is the case it is  precisely with the help of
turbulent mixing to connect surface water and intrusion, or maybe through convective instabilities
but the intrusion is within stable water column. Therefore  you may  rather insist on isopycnal
mixing as a possible missing process to explain dilution

**R:** We thank the reviewer for the constructive comment. Solar radiation and evaporation-
precipitation budgets were included to describe their broader influence on temperature and salinity
in the water column. However, as the reviewer correctly points out, their impact is primarily limited
to near-surface layers. Because of thi, the sentence was revised to focus on processes more relevant
to the depth range of the intrusion, emphasizing the role of isopycnal mixing as a key mechanism
for dilution.